# Genetic diversity and evolution of Hantaan virus in China and its neighbors

**Naizhe Li**[1], **Aqian Li**[1], **Yang Liu**[1], **Wei Wu**[1], **Chuan Li**[1], **Dongyang Yu**[2], **Yu Zhu**[2], **Jiandong Li**[1], **Dexin Li**[1], **Shiwen Wang**[1,3]*, **Mifang Liang**[1,3]*

**1** Key Laboratory of Medical Virology and Viral Diseases, Ministry of Health of People's Republic of China, National Institute for Viral Disease Control and Prevention, Chinese Center for Disease Control and Prevention, Beijing, China, **2** Department of Microbiology, Anhui Medical University, Hefei, China, **3** China CDC-WIV Joint Research Center for Emerging Diseases and Biosafety, Center for Biosafety Mega-Science, Chinese Academy of Sciences, Wuhan, P. R. China

* wangsw@ivdc.chinacdc.cn (SW); liangmf@ivdc.chinacdc.cn (ML)

**Data Availability Statement:** All sequences are available from the NCBI and ViPR database (S2 Table).

**Funding:** This work was supported by National Science and Technology Major Project of China

## Abstract

### Background

Hantaan virus (HTNV; family Hantaviridae, order Bunyavirales) causes hemorrhagic fever with renal syndrome (HFRS), which has raised serious concerns in Eurasia, especially in China, Russia, and South Korea. Previous studies reported genetic diversity and phylogenetic features of HTNV in different parts of China, but the analyses from the holistic perspective are rare.

### Methodology and principal findings

To better understand HTNV genetic diversity and gene evolution, we analyzed all available complete sequences derived from the small (S) and medium (M) segments with bioinformatic tools. Eleven phylogenetic groups were defined and showed geographic clustering; 42 significant amino acid variant sites were found, and 19 of them were located in immune epitopes; nine recombinant events and eight reassortments with highly divergent sequences were found and analyzed. We found that sequences from Guizhou showed high genetic divergence, contributing to multiple lineages of the phylogenetic tree and also to the recombination and reassortment events. Bayesian stochastic search variable selection analysis revealed that Heilongjiang, Shaanxi, and Guizhou played important roles in HTNV evolution and migration; the virus may originate from Zhejiang Province in the eastern part of China; and the virus population size expanded from the 1980s to 1990s.

### Conclusions/Significance

These findings revealed the original and evolutionary features of HTNV, which will help to illustrate hantavirus epidemic trends, thus aiding in disease control and prevention.

(2018ZX10711001). The funders had no role in study design, data collection and analysis, decision to publish, or preparation of the manuscript.

**Competing interests:** The authors have declared that no competing interests exist.

## Author summary

Hemorrhagic fever with renal syndrome (HFRS) and hantavirus pulmonary syndrome are endemic zoonotic infectious diseases caused by hantaviruses, which have threatened public health worldwide for decades. However, our knowledge about the emergence and evolution of HTNV still need to be improved. To get more information about HTNV genetic diversity and phylogenetic features with a holistic perspective, we investigated the genetic diversity and spatial distribution of HTNV using all available whole-genome sequences of small (S) and medium (M) segments collected from 18 different regions and using a larger timescale; 11 phylogenetic groups were defined. The sequences showed geographic clustering, and divergence with recombinant or reassortment events occurred. Using the Bayesian stochastic search variable selection method, geographic origins and migration patterns of HTNV epidemics were deduced. Our data provided important biological information to better understand hantavirus evolution, transmission, and epidemics.

## Introduction

Hantaviruses have gained worldwide attention as etiological agents of emerging zoonotic diseases—namely, hemorrhagic fever with renal syndrome (HFRS) in Eurasia and hantavirus cardiopulmonary syndrome in the Americas—with fatality rates ranging from <10% up to 60% [1]. Among all countries, China is the most seriously affected and accounts for over 90% of the total HFRS cases around the world [2, 3]. It has been reported that the HFRS death rate in China was 2.89% during the years 1950–2014 [4]. Although a declining HFRS trend has been observed at a global scale in China, there still exist certain local regions that continue to display increasing HFRS trends [5]. However, the causative agent of the disease remained unknown until the early 1970s, when Lee and colleagues reported on Hantaan virus (HTNV) present in the lungs of its natural reservoir, the striped field mouse (*Apodemus agrarius*) [6].

HTNV, as one of the pathogenic hantaviruses of HFRS, is a member of genus *Orthohantavirus*, family Hantaviridae, in the order Bunyavirales. The virus genome consists of three separate segments of negative-stranded RNA, referred to as small (S), medium (M), and large (L) segments, which encode nucleocapsid protein (NP), two envelope glycoproteins (Gn and Gc), and viral RNA-dependent RNA polymerase, respectively [7]. Recent studies [8–10] reported genetic diversity and phylogenetic features of HTNV in different parts of China, but the analyses from the holistic perspective are rare. Also, the mechanisms underlying the emergence and evolution of HTNV are poorly understood. Understanding the phylogenetic factors contributing to HTNV transmission has important implications for our understanding of its epidemiology and provides insights about surveillance as well as outbreak predictions and preparation.

In this study, we focused on the genetic diversity and evolutionary history of HTNV. Whole-genome sequences were analyzed with bioinformatic tools. The results revealed the phylogenetic relationships among HTNV strains with both Bayesian and maximum-likelihood (ML) methods. To deduce geographic origins and migration patterns of HTNV epidemics, Bayesian evolutionary analysis sampling trees (BEAST) software was used in this study. Furthermore, the recombination and reassortment events were also detected.

## Materials and methods

### Sequence data set

All the complete S gene and M gene sequences of *Hantaan orthohantavirus* deposited until June 2019 were collected from Virus Pathogen Database and Analysis Resource (ViPR, www.

viprbrc.org). Only one sequence was retained for the identity sequences with the same strain names. It should be mentioned that sequences from Shandong Province (accession no. KY639536-KY639711) were not defined as HTNV by both ViPR and Genebank but can be organized in *H. orthohantavirus* (see below). These sequences were still analyzed in this study. All the sequences were aligned using MAFFT version 7 [11] with default settings followed by manual refinement. The coding sequences were retained and used for the following analyses [12].

## Recombination detection

As recombination seriously affects phylogenetic inference, the whole data set was tested for the presence of recombination signals using RDP4 [13]. RDP, GENECONV, Maximum Chi-squared, Chimaera, Bootscan, Sister Scanning, and 3Seq were used only on events with *P* values < 0.01 that were confirmed by four or more methods, which were considered recombination events.

## Phylogenetic analyses and amino acid substitution analyses

It is reported that recombination sequences will bias the shape of the inferred phylogenetic tree, the branch length, and reconstruction of ancestral sequences [13–15]. To reduce the potential bias, the recombination segments were first removed from the data sets. The phylogenetic relationships of the complete S and M gene sequences were estimated using a Bayesian Markov chain Monte Carlo (MCMC) method as implemented in MrBayes v3.2.2 [16] and an ML phylogenetic inference as implemented in IQ-TREE v1.6.8 [17]. Dobrava virus, Puumala virus (PUUV), Sin Nombre virus, and Thottapalayam virus were designated as the out-group.

For Bayesian MCMC analysis, the suitability of substitution models for our data sets were assessed using jModelTest v2.1.10 [18], which performed a statistical model selection procedure based on the Akaike information criterion. It identified the best-fitting substitution model GTR+$\Gamma$ for both data sets of complete S and M gene sequences. The MCMC settings consisted of two simultaneous, independent runs with four chains each, which were run for 20 million generations and sampled every 200 generations with a 25% burn-in.

For the ML analysis, the best-fitting nucleotide substitution model that minimizes the Bayesian information criterion score was selected by ModelFinder [19], implemented in IQ-TREE. An ML tree was constructed using the best-fitting nucleotide substitution model, and statistical robustness of the branching order within the tree was assessed using ultrafast bootstrap support values [20] (1,000 replicates) and the SH-like approximate likelihood ratio test [21] (SH-aLRT, 5,000 replicates). The trees were visualized in FigTree v1.4.4 (http://tree.bio.ed.ac.uk/software/figtree/).

The Metadata-Driven Comparative Analysis Tool, supplied by ViPR, was used to analyze the variation of amino acid between the different phylogenetic groups, hosts, and sample collection years. The *P* value threshold was set to 0.05. If a specific amino acid exists only in one species or one group but not in other species or groups, this amino acid is considered as a "significant amino acid" marker (or synapomorphy).

## Natural selection analysis

Natural selection analysis was assessed using the Datamonkey web-based facility [22]. Site-specific selection pressure was assessed using four varying methods, including single-likelihood ancestor counting (SLAC), fixed-effects likelihood (FEL), mixed-effects model evolution (MEME), and fast unconstrained Bayesian approximation (FUBAR) [23–25]. Significance levels were set to $P < 0.05$, and posterior probability (PP) > 0.9. The SLAC, FEL, and MEME

methods were used to estimate the nonsynonymous to synonymous (dN/dS) ratio, which represents the differential effect of natural selection on these two types of mutations. The lower dN/dS values indicate stronger negative selection against amino acid change. To examine whether episodic diversifying selection had occurred on individual branches, we utilized the adaptive Branch-Site Random Effects Likelihood (aBSREL) [26] method, which is also available on the Datamonkey web server.

### Coalescent and evolution analyses

Complete S gene sequences were used to deduce the evolutionary history of HTNV. To avoid potential biases due to sampling heterogeneity, the data set was reduced using CD-HIT [27] by clustering together sequences with a nucleotide sequence identity threshold of 99%. Temporal evolutionary signal in the ML tree was evaluated by TempEst v1.5.1 [28], which plots sample collection dates against root-to-tip genetic distances obtained from the ML phylogeny tree.

The most recent common ancestor (tMRCA), substitution rates, and evolutionary history were estimated using a Bayesian serial coalescent approach implemented in BEAST v1.10.4 [29]. jModelTest v2.1.10 was used to determine the model of nucleotide substitution that best fit the data set, and the data set was subsequently run using GTR+I+$\Gamma$. Both strict and relaxed (uncorrelated exponential and uncorrelated lognormal) molecular clocks were used for the analysis, combined with different tree priors (constant population size, GMRF Bayesian Skyride, and Bayesian skyline were used in this study). Timescale was inferred using an informative substitution rate prior (a lognormal distribution with 95% of the density lying between $1 \times 10^{-4}$ and $1 \times 10^{-3}$ substitutions per site per year) previously estimated for the N gene of rodent-borne hantavirus [30]. Models were compared pairwise by estimating the log marginal likelihood via path sampling and stepping stone analysis [31–33]. For each model, an MCMC was run for 100 million generations, with sampling every 10,000 steps. All other priors were left on their defaults. Posterior probabilities were calculated using the program Tracer1.7 after 10% burn-in. The results were accepted only if all the parameters have effective sample sizes over 200.

The selected molecular clock/demographic model (strict clock with Bayesian skyline prior in this study) was then used for the Bayesian phylogeographic inference based on the continuous-time Markov chain process over discrete sampling locations to estimate the diffusion rates among locations, with Bayesian stochastic search variable selection. The MCMC was run for 300 million generations, with sampling every 30,000 generations. Convergence was assessed by estimating the effective sampling dates using Tracer and accepting effective sample size values of 200 or more for all the parameters. A maximum clade credibility (MCC) tree with the phylogeographic reconstruction was selected from the posterior tree distribution after a 10% burn-in using the Tree Annotator v1.10 and was manipulated in FigTree v1.4.4. The migration routes were visualized using SPREAD3 [34]. Migration pathways were considered to be important when they yielded a Bayes factor greater than 15 and when the mean posterior value of the corresponding migration event was greater than 0.50. Bayes factors were interpreted according to the guidelines of Kass and Raftery [35].

## Results

### HTNV sequences data set

A total of 225 complete S gene sequences and 180 complete M gene sequences from 238 HTNV strains were contained in the data set. The sampling dates of the sequences ranged from 1976 to 2017. Most of the samples were collected during 2011–2015 (Table 1). According to the sample collection area, 18 regions were defined, including Russia, South Korea, and 16 provinces of China (Fig 1). More samples were collected from South Korea and Zhejiang, with 67 and 36 strains,

**Table 1. The sampling dates and host distribution in different regions.**

| Region | Total | Years | | | | | | | Host | | | | |
|---|---|---|---|---|---|---|---|---|---|---|---|---|---|
| | | ≤1990 | 1991–2000 | 2001–2005 | 2006–2010 | 2011–2015 | ≥2015 | UN | *Aa* | *Rn* | *Hs* | Others | UN |
| Anhui | 1 | | | | | | | 1 | | | 1 | | |
| Guangdong | 1 | | | | | | | 1 | | | | | 1 |
| Guizhou | 23 | 8 | | 7 | | | | 8 | 9 | 9 | 5 | | |
| Heilongjiang | 6 | | | | 2 | 3 | | 1 | 5 | | 1 | | |
| Hubei | 4 | 1 | | 1 | | 1 | | 1 | 2 | | 2 | | |
| Hunan | 1 | | | | 1 | | | | 1 | | | | |
| Jilin | 4 | | 1 | | 2 | | | 1 | 2 | | | | 2 |
| Jiangsu | 13 | | | | | 12 | | 1 | 4 | 2 | 5 | Mf:2 | |
| Jiangxi | 6 | | | | | 5 | | 1 | 2 | | | Mm:3 | 1 |
| Liaoning | 7 | | | | 7 | | | | | | 7 | | |
| Russia | 8 | | 1 | | | | | 7 | 7 | | 1 | | |
| Shaanxi | 22 | 2 | | | 9 | 3 | 8 | | 10 | 1 | 11 | | |
| Shandong | 14 | | | | | 14 | | | 14 | | | | |
| Sichuan | 2 | | | | | | | 2 | | | | Cr:1 | 1 |
| South Korea | 78 | 1 | | 11 | 14 | 29 | 12 | 11 | 61 | | 11 | | 6 |
| Tianjin | 1 | | | | | | | 1 | | | | | 1 |
| Yunnuan | 2 | | 1 | 1 | | | | | | | | Nc:1 | 1 |
| Zhejiang | 38 | | | | 28 | 8 | | 2 | 31 | | | Mf:5 | 2 |
| UN | 7 | | | | | | | 7 | | | 1 | Tt:1 | 5 |
| Total | 238 | 12 | 1 | 22 | 54 | 84 | 20 | 45 | 148 | 12 | 45 | 13 | 20 |

Abbreviations: *Aa*, *A. agrarius*; *Hs*, *H. sapiens*; *Mf*, *Mirotus fortis*; *Mm*, *Mus musculus*; *Nc*, *Nivienter confucianus*; *Rn*, *R. norvegicus*; *Tt*, *Tscherskia triton*; UN, unknown

respectively. Furthermore, Guizhou and Shaanxi also had a higher number of HTNV samples. The samples were collected from nine kinds of hosts, including *A. agrarius* (*Aa*), *A. peninsulae* (*Ap*), *Rattus norvegicus* (*Rn*), *Mus musculus* (*Mm*), *Nivienter confucianus* (*Nc*), *Tscherskia triton* (*Tt*), *Mirotus fortis* (*Mf*), *Crocidura russula* (*Cr*), and *Homo sapiens* (*Hs*). *A. agrarius*, the main reservoir host for HTNV, has the highest number of samples, with a total of 148 (62.2%) strains isolated. The sampling dates and geographic distribution of the sequences are shown in Fig 1.

## Recombination events detection

Recombination events were detected among all the 225 complete S gene sequences and 180 complete M gene sequences from 238 HTNV strains collected in this study. The RDP analysis suggested nine recombination events in six HTNV isolates, and all the recombination events occurred in the M gene segment (Table 2). Similar recombination events were detected in three isolates (CGAa31MP7, CGAa31P9, CGHu2). All three isolates were collected from 2004 to 2005 in Guizhou Province [8]. This would imply that these isolates have the same ancestor, which has experienced recombination previously. Although only five methods suggested the recombination event occurred in strain JN131026, the *P* values were all lower than 0.01. Considering the consensus score was higher than 0.6 (0.646), we defined it as a recombinant isolate. *P* values of the event tested in strain A16 were in the high credibility level.

## Phylogenetic analyses and amino acid substitution analyses

The recombination isolates were excluded first. Our phylogenetic analysis showed that 11 groups were defined with the S segment, each with a high degree of support (Fig 2). The

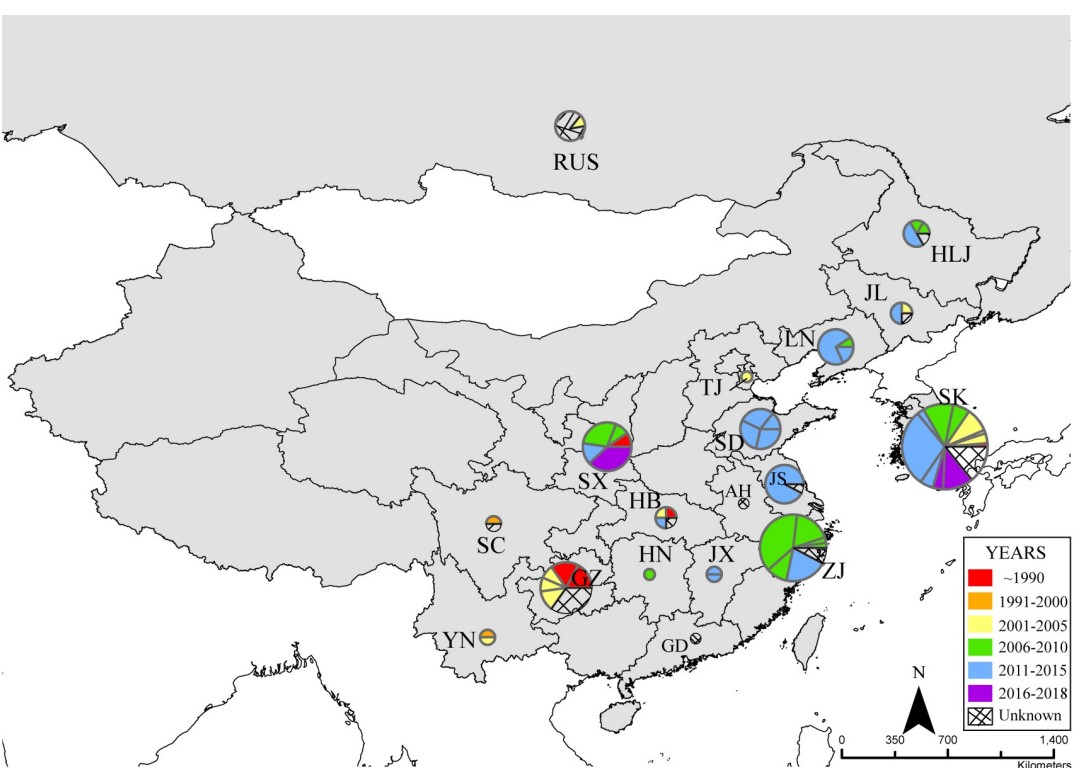

**Fig 1. Map showing the sampling dates and geographic distribution of HTNV isolates included in this study.** The geographic distribution of the HTNV isolates analyzed in this study. The size of a pie represents the number of isolates in the corresponding region. The pie slices are colored according to the sampling years. AH, Anhui; GD, Guangdong; GZ, Guizhou; HB, Hubei; HLJ, Heilongjiang; HN, Hunan; HTNV, Hantaan virus; JL, Jilin; JS, Jiangsu; JX, Jiangxi; LN, Liaoning; RUS, Russia; SC, Sichuan; SD, Shandong; SK, South Korea; SX, Shaanxi; TJ, Tianjin; YN, Yunnan; ZJ, Zhejiang. *Map source*: *Natural Earth* (https://www.naturalearthdata.com/).

isolates were mainly clustered together by region. The isolates from South Korea all belonged to group A. All the sequences from Jiangsu Province and one strain from Sichuan (S85_46) Province clustered in group A with sequences from South Korea. Group B was formed with sequences collected from Russia and the northeast of China. The isolates collected in Shaanxi were all clustered in group C, except one strain, 84FLi, which was isolated years ago in 1984. All the three isolates from Hubei constituted group D, and all the isolates from Shandong clustered as a subgroup in group E. The other subgroup of group E contains strains collected from Anhui, Hunan, Shaanxi, Sichuan, Guangdong, and Yunnan. The isolates collected from Zhejiang constitute group G and group H. Group J contains isolates all from Jiangxi. Group K was formed with three isolates from Russia. We noticed that the isolates collected from Guizhou Province were highly divergent. They dispersed in group B, C, F and I, indicating that HTNV has spread in Guizhou for a long period.

There were 10 distinct phylogroups in the M gene tree. The structure of the phylogenetic tree reconstructed by the M gene was identical to the S gene tree, except for seven isolates (CGRn15, CGAa2, CGAa75, CGRn2616, CGRn2618, CGRn45, CGHu3) derived from Guizhou Province. They were classified into group I, but in the S gene tree, they were included in group C. Additionally, isolate N8, collected form Jiangxi, was classified into group G in the M tree but group J in the S tree. This suggested that reassortment events occurred in these strains. The branching order within the lineages differed in S and M trees, which indicates the different

**Table 2. Recombination events detected in all HTNV sequences.**

| Events | Recombinant | Segment | Beginning | Ending | Major Parent | Minor Parent | Method(s) by which Breakpoint Was Detected in RDP4 |
|---|---|---|---|---|---|---|---|
| 1 | CGAa31MP7 | M | 1,756 | 1,916 | CGRni1 | CGRn15 | RDP/GENECONV/MaxChi/Chimaera/Bootscan/SisScan/3Seq |
| 2 | CGAa31MP7 | M | 2,315 | 3,402 | Q32 | CGRn2616 | RDP/GENECONV/MaxChi/Chimaera/Bootscan/SisScan/3Seq |
| 3 | CGAa31P9 | M | 1,756 | 1,916 | CGRni1 | CGRn15 | RDP/GENECONV/MaxChi/Chimaera/Bootscan/SisScan/3Seq |
| 4 | CGAa31P9 | M | 2,315 | 3,402 | Q32 | CGRn2616 | RDP/GENECONV/MaxChi/Chimaera/Bootscan/SisScan/3Seq |
| 5 | CGHu2 | M | 1,756 | 1,916 | CGRni1 | CGRn15 | RDP/GENECONV/MaxChi/Chimaera/Bootscan/SisScan/3Seq |
| 6 | CGHu2 | M | 2,315 | 3,402 | Q32 | CGRn2616 | RDP/GENECONV/MaxChi/Chimaera/Bootscan/SisScan/3Seq |
| 7 | CGRn93P8 | M | 2,386 | 3,402 | CGRn93MP8 | CGHu2 | RDP/GENECONV/MaxChi/Chimaera/Bootscan/SisScan/3Seq |
| 8 | JN131026 | M | 1,413 | 2,181 | JN131027 | 158577 | MaxChi/Chimaera/Bootscan/SisScan/3Seq |
| 9 | A16 | M | 2,214 | 3,402 | H8205 | SN7 | RDP/GENECONV/MaxChi/Chimaera/Bootscan/SisScan/3Seq |

Abbreviations: HTNV, Hantaan virus; M, medium

evolutionary history between the M and S segments. It should be clarified that all the branching orders were well supported.

We analyzed the variation of amino acids between the different phylogenetic groups. We noticed that five amino acid sites on NP gene and 37 on Gn/Gc gene were group specific and could be markers to distinguish different groups (Fig 3). No significant variant was found at the highly conserved pentapeptide motif WAASA, which is thought of as the cleavage site on glycoprotein precursor (GPC) polypeptide into the Gn and Gc transmembrane proteins [36]. To determine whether any of these amino acid sites were located in any known immune epitopes, we compared these significant positions against information about experimentally determined immune epitopes curated by the Immune Epitope Database. We found a total of 19 positions that were located in known immune epitopes (Table 3). No significant amino acid marker was found among different hosts or collected times.

## Natural selection analysis

The natural selection analysis showed that purifying selection plays a dominant role in HTNV evolution, with very low dN/dS values and an abundance of negatively selected sites (Table 4). The dN/dS values for Gn were the highest, but they still show little evidence of positive selection. In total, 10 positively selected sites in the Gn protein and 11 in the Gc protein were found by MEME. One and 14 positively selected sites in N protein were found by FUBAR and MEME, respectively. We noticed that none of the significant amino acid substitutions sites (Fig 3) were being subjected to positive selection. Evidence of episodic diversifying selection was detected in two branches (CGRn53, HoJo) in the phylogeny of M segment by aBSREL.

## Coalescent analyses and evolution of HTNV

To avoid potential biases in the phylogeographic reconstructions due to sampling heterogeneity, we obtained a "nonredundant" subset including 51 clusters. In order to make our data set represent all 15 regions (there was no sequence with exact collected year from Anhui, Sichuan, and Guangdong), isolate E142, Hunan03, and JS10 were added to the data set. Temporal evolutionary signal analyses showed a positive correlation between genetic divergence and sampling time (S1 Fig). It should be mentioned that though the data set we used showed positive correlation by root-to-tip analyses, it failed to pass the date-randomization test (DRT, S2 Fig) [37]. As DRT is stricter than root-to-tip analyses in clock signal evaluation, we think our data set still shows clock signal, which is not so strong. The strict clock with Bayesian skyline prior

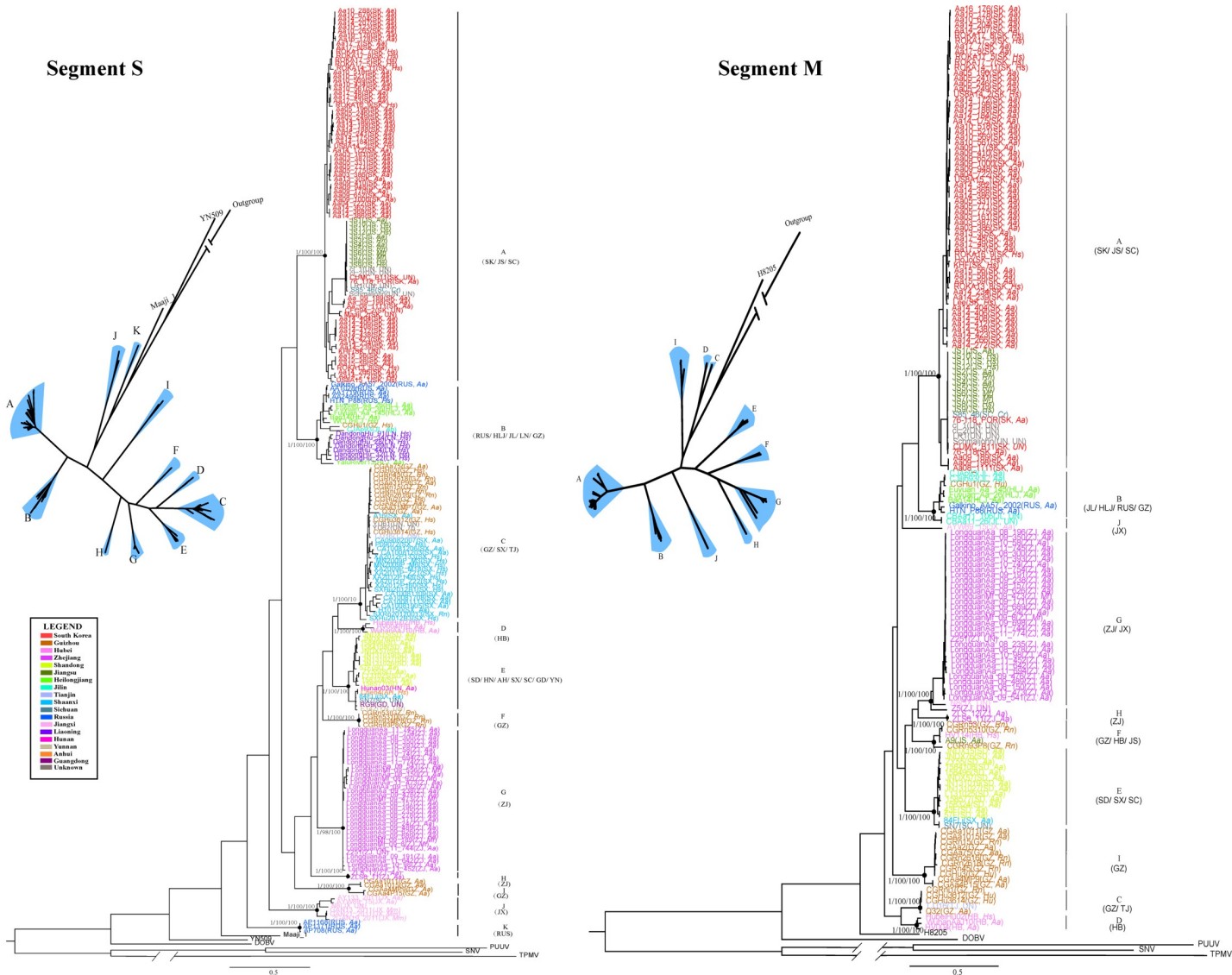

**Fig 2. Phylogenetic trees of S and M genes reconstructed by MrBayes.** Posterior node probabilities/SH-like approximate likelihood values/ultrafast bootstrap values for major nodes (black circles) are shown. Split network analysis of HTNV was also shown. The region and host features for each sample were labeled after the strains' names. The brackets below the group name indicated the regions associated to each group. AH, Anhui; GD, Guangdong; GZ, Guizhou; HB, Hubei; HLJ, Heilongjiang; HN, Hunan; HTNV, Hantaan virus; JL, Jilin; JS, Jiangsu; JX, Jiangxi; LN, Liaoning; M, medium; RUS, Russia; S, small; SC, Sichuan; SD, Shandong; SK, South Korea; SX, Shaanxi; TJ, Tianjin; YN, Yunnan; ZJ, Zhejiang.

model yielded a higher log marginal likelihood than the others, indicating the best-fit model for our data set (S1 Table).

The results of our Bayesian phylogenetic analysis showed that HTNV probably first emerged in Zhejiang Province of China (root PP = 0.28), with a most recent common ancestor in 1214 (95% credibility interval 590–1592; Fig 4A). The root PP of Jiangxi, Shaanxi, and Guizhou were also in higher levels. The MCC tree of HTNV obviously showed two separate clusters. One was composed of isolates from the northeast of China and its surroundings, and the other was composed of isolates from the south and middle of China. These indicated that HTNV has been spreading in China for a long time. Reconstruction of the demographic history using the Bayesian skyline plot revealed that the HTNV population size was relatively

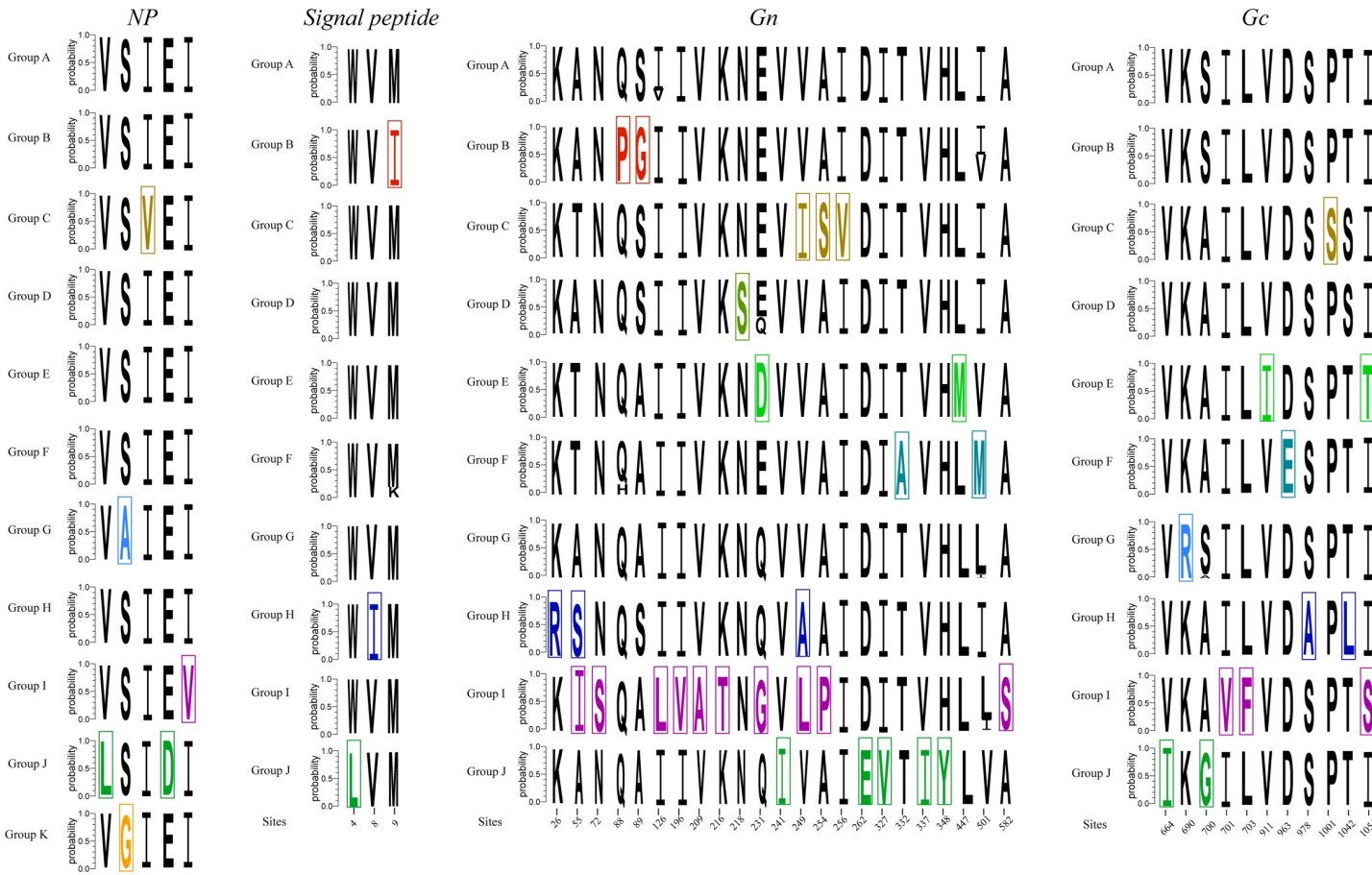

**Fig 3. The "significant amino acid marker" of each group.** The specific amino acid sites of each group are marked by different colors. NP, nucleocapsid protein.

constant from 1500 to the 1970s, expanded between the 1980s and 1990s, and then stayed steady from 2000 until now (Fig 3B). The mean substitution rate of $2.0185 \times 10^{-4}$ substitution per site per year, with a 95% highest posterior density (HPD) that ranged from $9.3435 \times 10^{-5}$ to $3.2011 \times 10^{-4}$, was estimated.

The spatiotemporal linkage of HTNV is shown in Fig 5. Significant location transitions were found mainly in two regions, the northeast and the middle of China (Fig 5A). This indicated that Heilongjiang may be the radiation center of the northeast regions of China and the surrounding areas. Significant migration events were found from Heilongjiang to Liaoning and the Far East of Russia. The migration from Heilongjiang to Khabarovsk was detected at a high credibility level (Bayes factor = 163.0). Shaanxi was assumed as the radiation center of the middle of China. Virus migration from Shaanxi to Tianjin, Yunnan, Hunan, and Guizhou was observed. The number of observed state changes pinpointed Shaanxi as the main source of HTNV epidemics in China, at least in the middle of China, and Heilongjiang as the dominating source of HTNV in northeast of China (Fig 5B).

## Discussion

In this study, we used the whole-genome sequences of segment S and M obtained from ViPR to analyze the genetic diversity and evolution of HTNV. Our analyses showed that HTNV can

**Table 3. The significant sequence variations' locations on immune epitopes.**

| Protein | Variation Site | Epitopes | | | | | |
|---|---|---|---|---|---|---|---|
| | | Epitope ID | Description | Amino Acid Change | Starting Position | Ending Position | B-/T-Cell Epitopes |
| NP | 179 | 164302 | NGIRKPKHLY**V**SLPN | V→L | 169 | 183 | T |
| | 296 | 807 | AEAAGCSM**I** | I→V | 288 | 296 | – |
| | 334 | 17746 | FS**I**LQDMRNTIMASK | I→V | 332 | 346 | T |
| | | 149047 | FS**I**LQDMRNTI | | 332 | 342 | T |
| Signal peptide | 8, 9 | 742143 | **VM**ASLVWPV | VM→IM/VI | 8 | 16 | T |
| Gn | 209 | 567849 | I**V**CFFVAV | V→A | 208 | 215 | T |
| | 216 | 77254 | **K216**, G217 | K→T | | | B |
| | 218 | 77257 | **N218**, L415 | N→S | | | B |
| | 501 | 568239 | PA**I**TFIIL | I→M,V,L | 499 | 506 | T |
| Gc | 664 | 743275 | G**V**GSVPMHTDLELDF | V→I | 663 | 677 | T |
| | 690 | 743113 | FSLTSSSKYTYRR**K**L | K→R | 677 | 691 | T |
| | | 743694 | KYTYRR**K**LTNPLEEA | | 684 | 698 | T |
| | 700, 701, 703 | 743887 | LTNPLEEAQ**S**IDLHI | S700G/A, I701V, L703F | 691 | 705 | T |
| | | 742696 | AQ**S**IDLHIEIEEQTI | | 698 | 712 | T |
| | 963 | 742777 | DHINILVTKDIDF**D**N | D→E | 950 | 964 | T |
| | | 40611 | LVTKDIDF**D** | | 955 | 963 | B |
| | 978 | 744025 | NLGENPCKIGLQTS**S** | S→A | 964 | 978 | T |
| | | 743579 | KIGLQTS**S**IEGAWGS | | 971 | 985 | T |
| | | 744567 | **S**IEGAWGSGVGFTLT | | 978 | 992 | T |
| | 1,001 | 744698 | TCLVSLTEC**P**TFLTS | P→S | 992 | 1,006 | T |
| | | 59502 | SLTEC**P**TFL | | 996 | 1,004 | T |
| | | 742922 | EC**P**TFLTSIKACDKA | | 999 | 1,013 | T |
| | 1,042 | 743186 | GKGGHSGS**T**FRCCHG | T→L/S | 1,034 | 1,048 | T |
| | | 744680 | S**T**FRCCHGEDCSQIG | | 1,041 | 1,055 | T |
| | 1,054 | 743149 | GEDCSQ**I**GLHAAAPH | I→S/T | 1,048 | 1,062 | T |

Abbreviations: NP, nucleocapsid protein

be divided into 11 groups. We named the groups based on the S tree for NP gene, which is more stable and has more sequences available. The isolates were geographically clustered. Sequences obtained from the same geographic area clustered together. It is worth noting that sequences from Guizhou were more diverse. We will discuss this later.

A total of 42 significant amid acid variant sites were found among the different groups. More significant variant sites and greater nonsynonymous variations were found for Gn than

**Table 4. Summary of selection pressures and codon sites under positive selection.**

| Gene | SLAC | | | FEL | | | FUBAR | | MEME | | |
|---|---|---|---|---|---|---|---|---|---|---|---|
| | dN/dS | N | Sites | dN/dS | N | Sites | N | Sites | dN/dS | N | Sites |
| NP | 0.0339 | 0 | – | 0.0341 | 0 | – | 1 | 43 | 0.0341 | 14 | 23, 32, 33, 50, 313, 326, 332, 396, 397, 398, 399, 401, 403, 405 |
| Gn | 0.0520 | 0 | – | 0.0450 | 0 | – | 0 | – | 0.0450 | 10 | 49, 53, 208, 213, 224, 275, 418, 520, 528, 548 |
| Gc | 0.0305 | 0 | – | 0.0242 | 0 | – | 0 | – | 0.0242 | 11 | 675, 724, 725, 780, 766, 860, 896, 925, 992, 1,045, 1,105 |

"N" refers to the number of positively selected amino acids, and "Sites" refers to amino acids sites that were found to be under positive selection.

Abbreviations: dN/dS, ratio of nonsynonymous to synonymous; FEL, fixed-effects likelihood; FUBAR, fast unconstrained Bayesian approximation; MEME, mixed-effects model evolution; NP, nucleocapsid protein; SLAC, single-likelihood ancestor counting

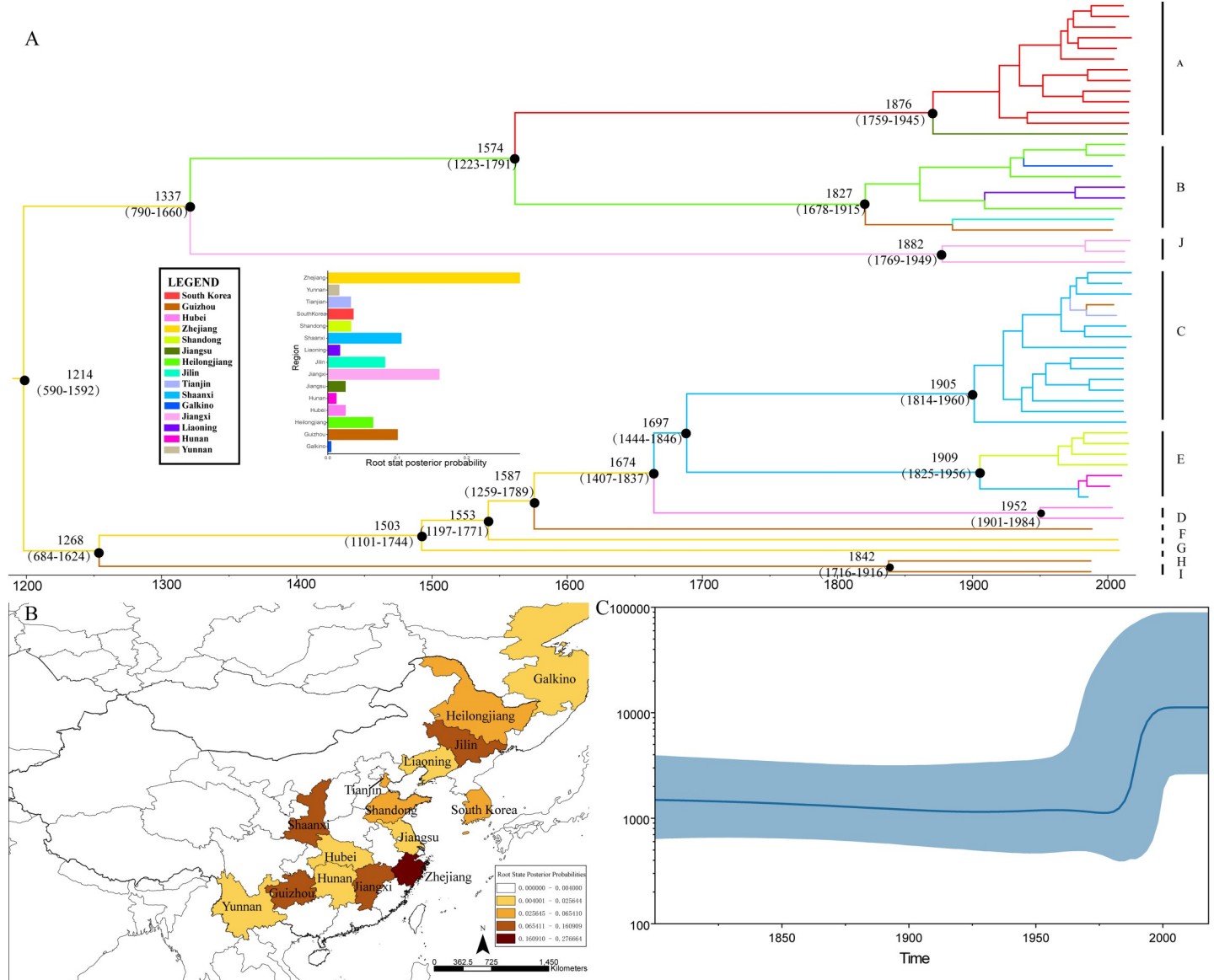

**Fig 4. The MCC tree, root state posterior probabilities, and effective population sizes of HTNV.** (A) MCC tree showing the evolutionary relationships and timescale of HTNV. Branch colors denote inferred location states, as shown in the legend. The most recent common ancestor and 95% credibility intervals are shown near the node. The root state posterior probabilities for the different regions were shown in the histogram. (B) A map with each region colored by the root state posterior probability. (C) Bayesian skyline plot showing population size through time for HTNV. Highlighted areas correspond to 95% HPD intervals. HPD, highest posterior density; HTNV, Hantaan virus; MCC, maximum clade credibility. *Map source*: *Natural Earth* (https://www.naturalearthdata.com/).

Gc, which is consistent with membrane-distal localization and supported the assumption that Gn was subjected to selective pressure of the humoral immune response [38]. But selection analyses showed none of these sites were being subjected to positive selection. We noticed more significant variants in group I. Isolates contained in group I were all from Guizhou. The collection dates were from the 1980s to 2000s. This also indicated high genetic diversity of HTNV in Guizhou. In total, 19 significant amid acid variant positions were found located at 25 known immune epitopes. Three of the epitopes are B-cell epitopes, and 21 are T-cell epitopes. It should be mentioned that epitope 807 was predicted by bioinformatics approaches and needs to be validated further. Liang and colleagues [39] reported peptide

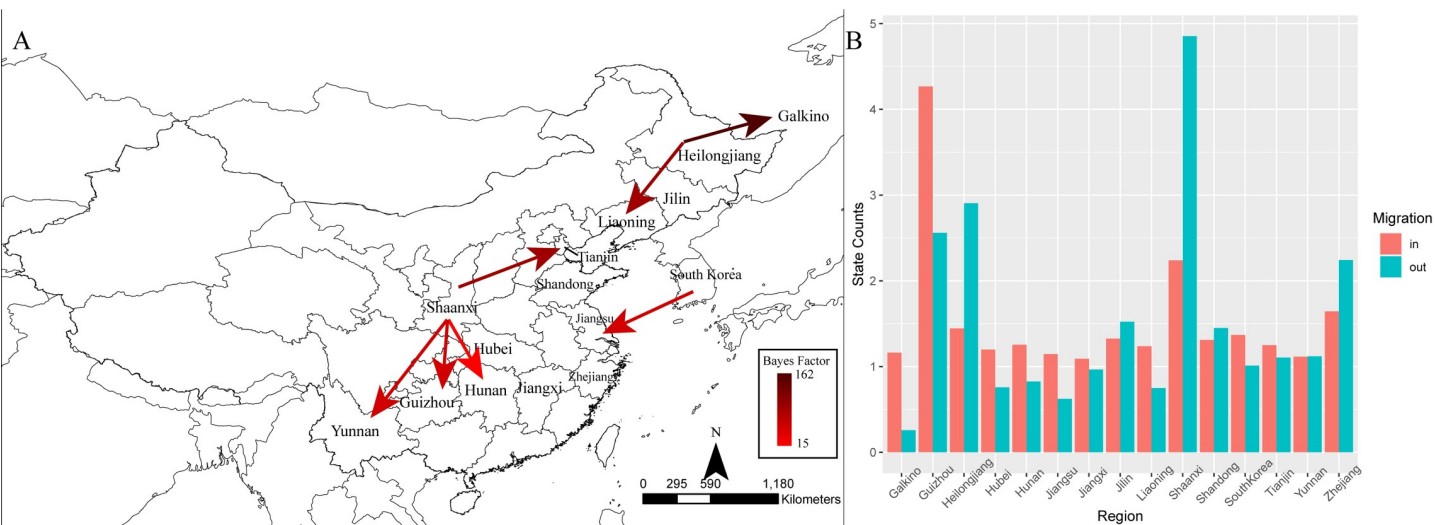

**Fig 5. Reconstructed phylogeographic linkage and the total number of location state transitions of HTNV.** (A) The phylogeographic linkage constructed reconstruction of the HTNV using BEAST. Color of lines represents the Bayes factor of migration between two regions. The map was created by SpreadD3 software, and the geographic data were provided by Natural Earth (https://www.naturalearthdata.com/). (B) Histogram of the total number of location state transitions inferred from the HTNV data set. BEAST, Bayesian evolutionary analysis sampling trees; HTNV, Hantaan virus.

437-GQRKVILTKLVIGQ-451 (although the epitope is not recorded in Immune Epitope Database), which showed high antibody binding and replacement at any position of the sequence LTKTLVIGQ (amino acid 443–451), leading to a substantial decrease in reactivity. We found a mutation L447M in group E, which contains isolates collected from Shandong, Shaanxi, and Sichuan. It is also reported [40] that amino acid exchanges of epitope (954-LVTKDIDFD-963) could lead to a loss in reactivity of binding with recombinant human antibodies. We found a mutation, D963E, in group F, which contains isolates from Guizhou, Jiangsu, and Hubei. This might be a clue to the possibility of immune escape in these regions. We found an epitope (ID 742143) located on signal peptide, which showed high binding affinity to HLA-A*0201 molecules and frequencies of epitope-specific cytotoxic T lymphocytes in the peripheral blood mononuclear cells of patients with HFRS and could induce CD8+ T-cell responses to inhibit HTNV replication [41]. Experimentation is needed to determine whether peptides containing these residues can confer escape or can be used to develop serotype-specific reagents for serology-based diagnostics. No significant amino acid marker was found among different hosts and periods, indicating that geographic region is the main factor affecting the genetic diversity of HTNV. The topologies between the M and S segments are different. We inferred that this was due to the different evolutionary history, which was in accordance with others' assumptions [42, 43], but this still needs to be validated in the future.

Our analyses showed that both reassortment and recombination play important roles in HTNV evolution. A number of studies have revealed that genetic reassortment can occur naturally or experimentally between the members of the Bunyavirales [44–49]. Seven of eight reassortment events occurred in M segments found in our study are consistent with Zou and colleagues [8]. We found nine recombination events in our study. All the recombination events occurred in segment M. The reassortment and recombination events found in Gn/Gc envelope glycoproteins were consistent with their functional roles in the viral escape from immunological responses. It should be noted that recombinant A16, isolated in Shaanxi, has both a major parent, H8205, and a minor parent, SN7. Strain H8205 has been classified into Amur virus, which indicates that recombination occurred across different species in

hantavirus. Strain SN7 was isolated in Sichuan, which implies the relationship of the evolution of HTNV between Shaanxi and Sichuan. Evidence of recombination has also been reported for *Tula orthohantavirus* [50] and *Puumala orthohantavirus* (PUUV) [51]; even the recombination in hantavirus is rare [52].

We investigated the evolution and migration of HTNV in China and its surrounding countries using Bayesian phylogeographic inference. Our phylogenetic analyses placed the root of the tree for HTNV in Zhejiang with strong support (Fig 3B). But no migration event was found from Zhejiang to other regions. This may result from the lack of whole-genome data from Zhejiang and its surrounding areas. But the reassortment event of strain N8 indicates the correlation of the isolates in Zhejiang and Jiangxi. Markov jump estimates between different locations pinpointed Heilongjiang as an important source of HTNV epidemics in northeast China and the Far East of Russia. This finding may explain the high level of epidemics in these areas since 1931, when HFRS was first recognized in northeast China [53, 54], due to the migration from Heilongjiang to its surroundings. In a similar pattern, Shaanxi was recognized as the origin of HTNV spread in the middle of China. These can be explained by a scenario in which the virus was first introduced into these areas and then expanded there. But the transmission sources to Heilongjiang and Shaanxi are still unclear. The samples collected from South Korea have been studied before [55–57], so we do not discuss them too much here. But the migration of HTNV from South Korea to Jiangsu indicates the possibility of HTNV import from other countries.

Our analyses showed the high diversity of isolates from Guizhou Province. Phylogenetic analyses showed that isolates collected from Guizhou Province clustered in group B, C, F, and I. Seven of eight reassortment events and seven of nine recombination events occurred in Guizhou. We found the same recombination pattern in strain CGAa31MP7, CGAa31P9, and CGHu2. All of these indicate that different sources of HTNV were transmitted to Guizhou and evolved for a long time in this place [58, 59]. This assumption could be supported by our Markov jump estimate (Fig 5). But our Markov jump estimate as well as the phylogenetic correlation with other regions also indicated the dispersion from Guizhou to other areas, even though no significant migration was found by phylogeographic inference. Furthermore, the PP that Guizhou was the root location was high, so we cannot deny the possibility of Guizhou to be the place of HTNV's common ancestor. With its rich mountainous topography, Guizhou is more than 1,000 meters above sea level, and as much as 92.5% of the province's total area is characterized by mountains. This makes *A. agrarius*, the main host of HTNV, sympatric in rural resident areas more common. The Hengduan Mountains region was hypothesized to have played an important role in the evolutionary history of *Apodemus* since the Pleistocene era [60, 61]. The Yunnan–Guizhou Plateau, which connects and overlaps with the Hengduan Mountains region, is also thought to play an important role in HTNV evolution. But we only found acceptable migration from Shaanxi to Guizhou in our study. This may result from the lack of sequences in Guizhou. To reduce the bias, recombinant and reassortment isolates were removed before the phylogeographic analysis. We also got rid of similar sequences with identity more than 99%. Thus, limited sequences from Guizhou were used for evolutionary and phylogeographic analyses. More gene information should be collected in order to know more about the HTNV evolution and spread in this area.

The most recent common ancestor of HTNV was determined about more than 800 years ago in our study, which is much earlier than when the first patient with HFRS was reported, in the early 1930s [62]. However, HFRS-like disease was described in Chinese writings about 1,000 years ago [63], so we believe HTNV has spread in China for more than 1,000 years. The most common recent ancestor according to Ramsden and colleagues [64] existed 859 years before the present for all rodent hantaviruses estimated and 202 years before the present for

Murinae viruses, with a mean substitution rate for rodent hantavirus of $6.67 \times 10^{-4}$ substitutions per site per year with a 95% HPD that ranged from $3.86 \times 10^{-4}$ to $9.8 \times 10^{-4}$ substitutions per site per year. Partial sequences (275 nt) were used in this study, which can explain the difference in tMRCA estimation. A recent study [65] showed that the estimated rate of nucleotide substitutions for the N gene of all rodent-borne hantavirus was $6.8 \times 10^{-4}$ substitutions per site per year, which is similar to $6.67 \times 10^{-4}$ substitutions per site per year. Our results implied that HTNV evolved at a lower speed compared with the other rodent-borne hantaviruses. The estimated rate of nucleotide substitutions of Dobrava virus and PUUV, $3 \times 10^{-4}$ and $5.5 \times 10^{-4}$ substitutions per site per year, respectively, supports our hypothesis [30].

The relatively recent origin of HTNV apparently contradicts the virus–host codivergence theory. Previous estimates of evolutionary dynamics in hantaviruses were based on the critical assumption that the congruence between hantavirus and rodent phylogenies reflects codivergence between these two groups because of the divergence of the rodent genera *Mus* and *Rattus*, approximately 10−40 million years ago, which indicates a mean substitution rate in the range of $10^{-8}$ substitutions per site per year [51, 66, 67]. However, the observation of host–pathogen phylogenetic congruence does not necessarily indicate codivergence. Phylogenetic congruence between a parasite and its host can also arise from delayed cladogenesis, in which the parasite phylogeny tracks that of the host but without temporal association [68]. This could occur if hantaviruses largely evolve host associations by cross-species transmission and related species tend to live in the same area, in which case a pattern of strong host–pathogen phylogenetic congruence could be observed in the absence of codivergence. Our evolutionary rates were estimated directly from primary sequence data sampled at known dates so that they more closely reflect the evolutionary changes undergone by the virus, at least in the short term. And with a mean substitution rate that is closer to other RNA virus, we consider our results to be more reliable than codivergence theory.

It should be noted that the substitution rate and tMRCA of HTNV estimated is not precise and could be improved in the future. As we show in the Results, the data set used for the molecular clock rate and phylogeography analyses did not pass the DRT, which revealed that the molecular clock rate of the data set is potentially unreliable. Therefore, a more accurate substitution rate and tMRCA would be estimated in the future when more sequences with stronger temporal structure are available. Furthermore, the selection pressure analyses showed that purifying selection plays a dominated role in HTNV evolution. It is reported [69] that the presence of strong purifying selection can lead to an underestimation of branch lengths. In our study, we calculated the total branch length of the MCC tree and aBSREL tree using the *compute.brlen()* function [70] supplied by ape package in R (http://ape-package.ird.fr/). The results showed that the total branch length for the MCC tree is smaller than the aBSREL tree (8.604 to 14.698), indicating the underestimation of the substitution rate and tMRCA of HTNV.

Our demographic analyses revealed that the HTNV population had expanded from the 1980s to 1990s. Because HTNV is a kind of rodent-borne virus, the population size of HTNV is relevant to rodent populations. Previous studies have revealed that climatic factors can influence HFRS transmission through their effects on the reservoir host (mostly rodents of the family Muridae) and environmental conditions [71–73]. It is reported that global annual average temperatures increased by more than 1.2˚F (0.7˚C) from 1986 to 2016 relative to temperatures seen from 1901 to 1960 [74]. Global warming may affect rodent winter survival through winter temperatures by a complicated process, and it may also influence the magnitude of HFRS outbreaks through summer climatic conditions (both temperature and rainfall). Additionally, as global climate change accelerates, the amount of annual rainfall increased accurately [75]. Moreover, agriculture improved rapidly in China during the same period, leading to more available food, which increases the reproduction rates of rodents. The steady effective

population from 2000 until now has resulted from the successful strategy for HTNV prevention in China; however, it is worth noting that the uncertainty of demographic analyses increases toward the root of the genealogy, where population history is reconstructed from fewer lineages, and a majority of samples in our data set were collected in recent years. Thus, the population size of the viruses may not be precise, but the increase from the 1980s to 1990s and the constant size after 2000 could be used as a reference for HTNV demographic inference.

However, some caution must be taken when interpreting our results. We have estimated the age of genetic diversity based only on N gene sequences; the evolutionary rate of HTNV may be underestimated. The Gn/Gc envelope glycoproteins are more diverse and considered to evolve at a higher speed. Furthermore, there may be geographic biases due to the imbalance in the collection of sequences from different regions. Because we did the research using whole-genome sequences, partial sequences are not comparable; for instance, partial sequences from different research may be located in different regions of whole genome. The lack of whole-genome sequences in some regions may result in the lack of knowledge about HTNV migration. In addition, lack of sequences of L gene can be obtained now. More gene information should be collected and shared to solve these problems.

In conclusion, our study revealed 11 groups of HTNV and 43 significant amino acid markers for different groups which could be connected with different regions. We hope it could be an indicator of immune effect evaluation of vaccine in different regions. Both recombination and reassortment events can be detected in HTNV. The origin and migration of HTNV were also shown by our analyses. We found that Heilongjiang, Shaanxi, and Guizhou played important roles in HTNV evolution and migration. It is crucial to pay more attention to HFRS prevention and control in these areas. Because rodent populations and activity influence the spread and increase of HTNV, rodent prevention and control is an effective way to reduce the incidence of diseases, which has also been proved by other studies [76, 77]. And a steady effective population from 2000 until now indicates that this is a successful strategy for HTNV prevention. These data provide important insights for better understanding the genetic diversity and spatiotemporal dynamics of HTNV and would be useful for disease prevention and control.

## Supporting information

**S1 Fig. Root-to-tip regression analyses.**
(TIF)

**S2 Fig. Comparison of the substitution rates estimated using BEAST from both the original and 20 date-randomized data sets from the DRT.** BEAST, Bayesian evolutionary analysis sampling trees; DRT, date-randomization test.
(TIF)

**S1 Table. Log marginal likelihoods computed by path sampling and stepping stone sampling.**
(XLSX)

**S2 Table. Accession numbers of the sequences using in this study.**
(XLSX)

## Author Contributions

**Data curation:** Naizhe Li, Yang Liu, Wei Wu, Chuan Li, Dongyang Yu, Yu Zhu.

**Formal analysis:** Naizhe Li, Aqian Li.

**Investigation:** Mifang Liang.

**Methodology:** Naizhe Li, Aqian Li.

**Project administration:** Jiandong Li, Dexin Li, Shiwen Wang.

**Resources:** Naizhe Li.

**Software:** Naizhe Li.

**Supervision:** Jiandong Li, Dexin Li, Shiwen Wang, Mifang Liang.

**Validation:** Naizhe Li.

**Visualization:** Naizhe Li.

**Writing – original draft:** Naizhe Li.

**Writing – review & editing:** Naizhe Li, Aqian Li, Yang Liu, Wei Wu, Chuan Li, Dongyang Yu, Yu Zhu, Jiandong Li, Dexin Li, Shiwen Wang, Mifang Liang.

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
