## [Decision Letter · Decision Letter 0]

12 Apr 2020

Dear Liang,

Thank you very much for submitting your manuscript "Genetic diversity and evolution of Hantaan virus in China and its neighbors" for consideration at PLOS Neglected Tropical Diseases. As with all papers reviewed by the journal, your manuscript was reviewed by members of the editorial board and by several independent reviewers. In light of the reviews (below this email), we would like to invite the resubmission of a significantly-revised version that takes into account the reviewers' comments. 

We cannot make any decision about publication until we have seen the revised manuscript and your response to the reviewers' comments. Your revised manuscript is also likely to be sent to reviewers for further evaluation.

Sincerely,

Alain Kohl

Guest Editor

Waleed Al-Salem

Deputy Editor

Reviewer's Responses to Questions

**Key Review Criteria Required for Acceptance?**

**Methods**

-Are the objectives of the study clearly articulated with a clear testable hypothesis stated?

-Is the study design appropriate to address the stated objectives?

-Is the population clearly described and appropriate for the hypothesis being tested?

-Is the sample size sufficient to ensure adequate power to address the hypothesis being tested?

-Were correct statistical analysis used to support conclusions?

-Are there concerns about ethical or regulatory requirements being met?

Reviewer #1: The objectives of the study are clear, as this research is largely exploratory, hypothesises are not required.

The study design is appropriate to the stated objectives.

The source of viruses sequences are clearly described and appropriate.

The sample sizes are appropriate for an exploratory phylogenetic study such as this.

The correct statistical analyses were used throughout, though see the important comment on the priors in the results section.

No ethical concerns.

Notes on specific issues that should be addressed:

At line 180 it is noted that “ Only parameter estimates with ESS values exceeding 200 were accepted.” Does this mean that all parameter values had ESSes exceeding 200 or just that those used in the analysis did? Ideally all parameters should have ESSes exceeding that threshold not just those parameters of direct interest, because there is no explicit tracking of the mixing of the tree topology, which often mixes worse that everything else, and most other parameters are themselves conditional on that topology. If a significant number of parameters in the analyses did not have ESSes greater than 200, the analysis should be rerun. If all parameters did have ESSes greater than 200, that statement should be modified for clarity.

Reviewer #2: Objectives of the study are clear and the study design is in keeping with similar published analyses. Sample size is sufficient to address hypotheses being tested, however tests taking strong negative selection into account are needed to properly estimate branch lengths and therefore TMRCA analysis. Additionally, the substitution rate should be validated using a date-randomisation test and the temporal signal of the dataset, which the authors analysed using TempEST, should be shown in order to give an idea of how strong the clock signal is.

Reviewer #3: (No Response)

**Results**

-Does the analysis presented match the analysis plan?

-Are the results clearly and completely presented?

-Are the figures (Tables, Images) of sufficient quality for clarity?

Reviewer #1: The analysis presented matches the analysis plan.

Results are clearly presented.

Figures and tables are of sufficient quality for clarity, however, some changes are suggested below.

At line 262 it is noted that “The branching order within the lineages differed in S and M trees, which indicates the different evolutionary history between the M and S segments.” It would be useful to add a sentence clarifying if this difference in branching order between the S and M segments is well supported, or whether there is high topological uncertainty in the relevant regions.

The estimation of the substitution rate in this paper is concerning. The bounds estimated are “a 95% HPD that ranged from 1.0001 х 10-4 to 3.2076 х 10-4 . The lower bound is butting right up against the edge of the uniform prior placed on the parameter by the authors, which ends at 1 х 10-4. This is a strong indicator that the data supports a lower evolutionary rate than the prior is allowing. A looser prior (such as a lognormal with 95% of the density lying between 1 х 10-4 and 1 х 10-3) would have avoided this issue, while providing much the same prior information. The model should be rerun with such a looser prior to test the how this changes the estimated evolutionary rate unless the authors have a convincing argument that their assessed subjective probability of the the rate being below 1 х 10-4 genuinely is zero (though this seems unlikely).

Table 2 is unclear, as it is not actually stated what the amino acid change is in each epitope. A column showing something along the lines of “V -> I” should be added in order to make the table more user friendly.

Figure 3 is weirdly stretched and letters have different heights (the “I”s for example are disproportionately tall), this should be amended, so that at least all letters have the same height.

It would be useful to change pane B of figure 4 to a map of China with each province coloured by the posterior probability that it contained the root, so that the data can considered spatially, as it is discussed in the text (i.e. to see how highly supported areas are spatially co-located).

Reviewer #2: Data presently closely matches the analysis plan and the results are clearly laid out in a sensible and logical order. The authors note that isolates from Guizhou province were highly divergent and group in phylogroups B, C, E, F and I. They do not discuss the possibility of movement of HTNV between areas at this point, and whether it is possible that isolates which evolved in Guizhou province have been dispersed recently into other regions - this would tie together with figure 6 where the movement of HTNV is analysed.

Reviewer #3: (No Response)

**Conclusions**

-Are the conclusions supported by the data presented?

-Are the limitations of analysis clearly described?

-Do the authors discuss how these data can be helpful to advance our understanding of the topic under study?

-Is public health relevance addressed?

Reviewer #1: Conclusions are supported by the data presented.

The limitations of the analysis are clearly presented.

The authors discuss how these data advance our understanding of the topic under study.

Public health relevance is not directly addressed.

At line 379 the authors note: “Markov jump estimates between different locations pinpointed Heilongjiang as an important source of HTNV epidemics in northeast of China and far east of Russia. This finding may explain the high endemic in this area since 1931, when HFRS was first recognized in northeast of China.” It seems more likely that the presence of virus in Heilongjiang allowed it to become the source of the the HTNV epidemics in other regions rather than vice versa. This alternative possibility should be quickly addressed.

The authors should comment on how much of the estimated recent increase in the population size of the viruses estimated in their skyline analysis they believe is real and how much is due to the sampling of recent diversity being much more complete than historical diversity. If few lineages that were present historically survive, there is limited information to estimate the demography backwards in time, as much of the diversity that was present then is missing from a sample collected in the present era, an issue that may be exacerbated by the auto-correlative effect that is inbuilt in the skyline demographic prior which penalises changes in demography when the evidence for them is weak.

Reviewer #2: The conclusions of the authors are supported by the data however these conclusions could be reinforced by additional analyses as described below. The limitations of the data are clearly discussed by the authors and the public health relevance of HTNV and the study are discussed at length.

Reviewer #3: (No Response)

**Editorial and Data Presentation Modifications?**

Reviewer #1: The manuscript uses acronyms before they are defined in the text. The definition of an acronym should occur with the first use throughout. (e.g. HFRS at line 22)

The authors should note whether any priors for Bayesian analyses were modified from their default values beyond those mentioned in the text. If not, the statement “All other priors were left on their defaults” would aid in clarity.

Reviewer #2: Fig.1 is difficult to interpret - it may be more effective to convey the isolate sampling locations on a map, with pie charts depicting the years of isolations of samples with the size of the charts directly related to the number of samples from that region. It may also be more effectively communicated as a simple table. A map of the sampling locations would also be helpful when interpreting the ML trees presented in Fig.2, to help the reader determine whether geographically distant isolates also appear more or less divergent. Figure 2 is missing a distance bar showing the divergence of the isolates as substitutions/site/year. Figure 4 is presented well however the colour coding changes from part A to part B making interpretation of the root analysis more difficult. In figure 5 the Bayes factor of the movement may be better presented as a colour gradient as opposed to line width, with a legend clearly showing the gradient - this may be easier to interpret than line width.

Reviewer #3: (No Response)

**Summary and General Comments**

Reviewer #1: I think that this is a good study, providing important background to the evolution of Hantaan virus. The analysis was rigorous, and the results presented are very interesting. I however, cannot in good faith recommend publication until the Bayesian phytogeographic analysis repeated with a prior that does not have a hard bound at 1 х 10-4 for the reasons that I articulated in the Results section, if only because it is unclear to me what effect this assumption has on all the other phylogeographic and demographic results in the manuscript. 

I do not want the authors to take this as a damning criticism however, I think that broadly this is a very good piece of work, and once the effects and of this assumption are tested and my other comments are addressed, I will be very happy to recommend it for publication.

Reviewer #2: Study is well presented and the methods and bioinformatic approaches used by the authors are standard and well characterised. The conclusions reached by the authors are acceptable but could be validated further.

Authors identify amino acid substitutions present in phylogroups, and show that several of these are in known immune epitopes, however it would be informative to complement this with a selection analysis. This would determine whether any of these sites are being subjected to positive selection which would lend credence to the importance of the substitutions affecting known immune epitopes.

Additionally, analysis of selection in the dataset would be helpful to shed light on the level of negative selection working on the HTNV genomes. Previous analyses using smaller datasets have shown that HTNV is subjected to strong purifying selection so this larger dataset could corroborate this. More importantly, it has been shown that the presence of strong purifying selection can lead to an underestimation of branch lengths which, when investigating the tMRCA, can result in a spurious, recent estimation (Wertheim, J.O., Pond, K., L, S., 2011. Purifying Selection Can Obscure the Ancient Age of Viral Lineages. Mol Biol Evol 28, 3355–3365. https://doi.org/10.1093/molbev/msr170).

The authors admit "The relatively recent origin of HTNV apparently contradicts the viral-host codivergence theory", therefore it would be helpful to re-estimate a ML tree using the adaptive branch site random effects likelihood (aBS-REL) model in addition to the GTR+Γ model in order to compare the reaulting branch lengths and therefore determine the validity of the estimated tMRCA.

The authors investigate the evolutionary history of HTNV using BEAST, however they first analyse the clock signal of the data set using TempEST - it would be helpful to include this graph in the supplementary figure in order to show how strong the temporal signal is. Furthermore, the validity of the estimated substitution rate could be explored using a date-randomization test, in which the dates of sampling for each sequence are randomised to present a null model with which to compare substitution rates (Duchêne, S., Duchêne, D., Holmes, E.C., Ho, S.Y.W., 2015. The Performance of the Date-Randomization Test in Phylogenetic Analyses of Time-Structured Virus Data. Mol. Biol. Evol. 32, 1895–1906. https://doi.org/10.1093/molbev/msv056)

Reviewer #3: In their manuscript, Li et al present an analysis of the molecular diversity and spatio-temporal distribution of HTNV genomic sequences found in different provinces of China and nearby areas of South Korea and Russia. HTNV, hosted by rodents of the muridae family, is endemic in China and causes many human cases of HFRS. To get insight into the molecular diversity and evolutionary history of orthohantaviruses, the authors make use of S and M sequence datasets from 238 HTNV strains, collected from 1976 to 2017, available on the “ViPR” website. The authors applied many computer programs and maximum likelihood approaches to built phylogenetic trees, defining 11 different phylogroups of HTNV clustering in specific geographical areas and exhibiting specific amino acid substitutions. Then, by Bayesian phylogeographic inference and time scale studies, they identify migration patterns, population density and emergence, more than 800 years ago in the Zhejiang province, of the most recent common ancestor of HTNV in China. 

Major points.

The study is of great interest for epidemiologists and virologists to understand evolution of orthohantaviruses. It also concerns computer scientists and mathematician since it entirely relies on bio-informatics tools. More and more investigations using databases report on genetic diversity of HTNV in provinces of China or South Korea. The present study covers a larger area and time scale. However the reading of the manuscript is made difficult due to the use of many informatics programs abbreviated and listed in the results. Analyses of the results supporting the driven conclusions are not always clearly exposed. 

1- The detail of the sequences in the datasets used in the study and presented in the fig 1 as stack columns is very difficult to interpret. It would help having first a panel showing a map of the provinces from east China and neighbouring South Korea and Russia with spot illustrating a range for the number of collected sequences. A table summarizing the data of the number of sequences collected per region, host and years could also be added. Also in the corresponding result section there is no remark made on the distribution of the collected sequences. The authors should comment for instance on the higher number obtained from South Korea then Zhejang, the main hosts and also the few isolates from Rattus norvegicus and possible relationship between HTNV prevalence in the natural reservoir and human host in different regions.

2- The authors eliminated recombination and reassortment events from analysis to reduce bias but the potential impact it could have on the phylogenetic trees is not explained, although such events contribute to the evolution of orthohantaviruses. The group clustering of S an M sequences together with regions is clear. However, the many different colours used in the Fig.2, make the reading difficult. In addition to the colour legend, the authors should add, into brackets below the letters identifying the phylogroups, right side of the trees, the same abbreviations as used in fig.1, to indicate the regions associated to each group. Also the identification of 42 specific amino acid substitutions associated to the different groups is interesting. The authors could discuss more the relationship with the immune epitopes: - are they B or T cells epitopes: -the fact that most of the varying sites are found in Gn (26/42), but only 6 are located in a immune epitope in contrast to the 11/42 sites found in Gc among which as much as 10 positions belong to such epitopes; -etc.. 

3- The different ML approaches, phylogeographic linkage and state transition analyses lead to an impressive amount of information but still some aspects are quite intriguing and could be worth considering such as: -the origin of the common ancestor in Zhejang in 1198, not appearing as a radiation center, -the high genetic diversity of HTNV from Ghizou segregating in different phylogroups, explained by a long evolution history relating to topology but also a lack of sequences. Could Ghizou be the place of HTNV common ancestor in China This should be confronted to the study of Zou et al (8) proposing Ghizou as a radiation centre, -the different evolution history of M and S segment also associated to topology is not clear. 

4- Important issues to be addressed are whether and how a low number or incompleteness of sequences, as well as including or not other regions, could impact the results. The case of South Korea and phylogeographic linkage could be further discussed. The authors propose that their findings could help to understand hantavirus epidemics and to develop prevention strategies. They should discuss in which way and include similar recent approaches strengthening this idea (for instance, Zheng et al 2019 doi.org/10.1371/journal.pntd.0007148) 

Other points

English language should be revised and many typos and spelling mistakes corrected. Too many abbreviations are used and some are not defined (tMRCA lane 157, HPD lane 311). 

Authors should define what they mean by analyses done from a holistic perspective (lanes26, 53 and 83) as compared to traditional approach (lane 57).

At the end of the author summary, from lane 52, sentences are just redundant. Such repetition occurred several times as for instance in the discussion lanes 339-341 

The nomenclature should be corrected by referring to the last ICTV recommendations with italic and majuscules used accordingly and should be harmonised along the manuscript. For instance HTNV does not belong to the Bunyaviridae family (lane 48) but to the Hantaviridae family in the genus Orthohantavirus as written lane 77.

The colour code in Fig.4B defining the regions is different from the one in Fig.4A. This is confusing and this legend is just unnecessary and should be removed since the regions are clearly written on the left of the plot fig.4B. The authors could just draw black bars. In the case of Fig 5A the visualisation of the phylogeographic linkage could be better if diminishing the size of the character font.

PLOS authors have the option to publish the peer review history of their article (what does this mean?). If published, this will include your full peer review and any attached files.

Reviewer #1: No

Reviewer #2: Yes: Jordan J. Clark

Reviewer #3: No
---

## [Decision Letter · Decision Letter 1]

19 Jun 2020

Dear Liang,

Thank you very much for submitting your manuscript "Genetic diversity and evolution of Hantaan virus in China and its neighbors" for consideration at PLOS Neglected Tropical Diseases. As with all papers reviewed by the journal, your manuscript was reviewed by members of the editorial board and by several independent reviewers. The reviewers appreciated the attention to an important topic. Based on the reviews, we are likely to accept this manuscript for publication, providing that you modify the manuscript according to the review recommendations. 

Please take for the revision into account comments on BEAST analysis, check for consistency/errors and ensure the paper can be carefully re-read by a native English speaker.

Sincerely,

Alain Kohl

Guest Editor

Waleed Al-Salem

Deputy Editor

Please take for the revision into account comments on BEAST analysis, check for consistency/errors and ensure the paper can be carefully re-read by a native English speaker.

Reviewer's Responses to Questions

**Key Review Criteria Required for Acceptance?**

**Methods**

-Are the objectives of the study clearly articulated with a clear testable hypothesis stated?

-Is the study design appropriate to address the stated objectives?

-Is the population clearly described and appropriate for the hypothesis being tested?

-Is the sample size sufficient to ensure adequate power to address the hypothesis being tested?

-Were correct statistical analysis used to support conclusions?

-Are there concerns about ethical or regulatory requirements being met?

Reviewer #1: (No Response)

Reviewer #2: The authors have done very well to address the comments of the reviewers. With the exception of some minor points the manuscript is ready for publication.

Reviewer #3: (No Response)

**Results**

-Does the analysis presented match the analysis plan?

-Are the results clearly and completely presented?

-Are the figures (Tables, Images) of sufficient quality for clarity?

Reviewer #1: (No Response)

Reviewer #2: The authors have amended the figures exactly as recommended by the reviewers.

Reviewer #3: (No Response)

**Conclusions**

-Are the conclusions supported by the data presented?

-Are the limitations of analysis clearly described?

-Do the authors discuss how these data can be helpful to advance our understanding of the topic under study?

-Is public health relevance addressed?

Reviewer #1: (No Response)

Reviewer #2: The authors carried out the date randomisation test as requested, which revealed that the molecular clock rate of the dataset is potentially unreliable. This information should be presented before any discussion of the molecualr clock rate and phylogeography, as discussing results before revealing that they are unreliable may be misleading to readers. This would ideally be described on line 284. It would also be interesting for the authors to discuss the potential mechanism behind this unreliable clock rate.

The authors also carried out the selection analysis as requested, including using the aBS-REL method. As mentioned, strong purifying selection can lead to an underestimation of branch lengths, however the authors have not fully investigated this. In order to determine whether their branch lengths are under estaimated the total branch length of the BEAST tree, and the tree produced by the aBS-REL method must be calculated in R using the ape package (http://ape- package.ird.fr/). If the total branch length for the BEAST tree is smaller than the aBS-REL tree, then the purifying selection in the dataset results in an underestimation of branch lengths and therefore a spurious substitution rate and TMRCA.

Reviewer #3: (No Response)

**Editorial and Data Presentation Modifications?**

Reviewer #1: (No Response)

Reviewer #2: The authors have amended all previous editorial mistakes.

Reviewer #3: The authors addressed most of the points raised in my previous report and clarified many aspects of the results. They also modified the figures, as suggested, making them now more easily understandable and helpful.

I suggest that the authors correct a few inconsistencies in their manuscript:

- lane 190-192: since non explained abbreviations are used for the different hosts, in table 1 and figure 2, they should be introduced first after each species inside brackets.

- lane 194: the detail of the sequences are not appearing in fig 1 but in fig 2, also lane 197 the map shows a “geographic distribution” rather than ‘the localities of HTNV?”

- lane 234: it is written that isolates from Guizhou dispersed in group B,C, E, F and I, but neither N nor M sequences isolated from Guizhou province are found in group E as shown in fig 2.

- page 17: the new paragraph about natural selection is not clearly presented and it will be important to define in the text what is dN/dS used in the table 4.

- Lane 363: reassortment can occur between arthropod-borne viruses from the Bunyavirales order, but not only, since this has been described also for rodent-borne hantaviruses , as indicated in the references (44-48) mentioned by the authors.

- Lane 382: “ this finding may explain the high endemic(?) in these areas..” Do the authors mean “this finding might be explained by high-endemicity… or that this finding may explain the high level of epidemics in this area…” which is then redundant with the previous sentence (lane 381).

Also, there are still many spelling and syntax errors to be corrected in order to improve the writing.

**Summary and General Comments**

Reviewer #1: Given that I answered all these questions the first time around, I will ignore them this time. Just to say, I am happy with the way the authors have addressed the changes I requested, and am happy to now recommend this paper for publication. Below I will note some copy-editing changes that I noticed are required as I went back through the manuscript.

Line 85: "bio-information" should be "bioinformatic"

Line 189: "had higher number" should be "had a higher number"

Line 359: "others’ assumption" should be "others' assumptions"

Line 384/385: "Similar pattern" should be "In a similar pattern"

Line 389: "discuss too much here" should be "discuss them too much here" or similar

Line 400: "Furthermore, root state posterior probability of Guizhou was also in a high level." This is clunky. Something similar to "Furthermore, the posterior probability that Guizhou was the root location was high." would be better.

Reviewer #2: The authors have done very well to address the comments of all reviewers. The revised data presentation and revised BEAST analyses are of great benefit to the manuscript. Everything asked of them has been covered with the exception of two small points:

In figure S1, there are four outlier sequences isolated between 1980 and 1990 who's root-to-tip divergence does not match well with the tiem oftheir isolation. Have the authors tried excluding these fromt the analysis? These sequences may contribute to the unreliable clock rate.

As discussed above, total branhc lengths for the BEAST and aBS-REL tree must be calculated and comapred.

Reviewer #3: (No Response)

PLOS authors have the option to publish the peer review history of their article (what does this mean?). If published, this will include your full peer review and any attached files.

Reviewer #1: No

Reviewer #2: No

Reviewer #3: No
---

## [Editor Report · Decision Letter 2]

8 Jul 2020

Dear Liang,

We are pleased to inform you that your manuscript 'Genetic diversity and evolution of Hantaan virus in China and its neighbors' has been provisionally accepted for publication in PLOS Neglected Tropical Diseases.

Best regards,

Alain Kohl

Guest Editor

Waleed Al-Salem

Deputy Editor

---

## [Editor Report · Acceptance letter]

23 Jul 2020

Dear Liang,

We are delighted to inform you that your manuscript, "Genetic diversity and evolution of Hantaan virus in China and its neighbors," has been formally accepted for publication in PLOS Neglected Tropical Diseases.

Best regards,

Shaden Kamhawi

co-Editor-in-Chief

Paul Brindley

co-Editor-in-Chief
